Methods

# TAF-ChIP: an ultra-low input approach for genome-wide chromatin immunoprecipitation assay

Junaid Akhtar[1], Piyush More[2,*], Steffen Albrecht[5,*], Federico Marini[3,4], Waldemar Kaiser[1], Apurva Kulkarni[1], Leszek Wojnowski[2], Jean-Fred Fontaine[5], Miguel A Andrade-Navarro[5], Marion Silies[1], Christian Berger[1]

Chromatin immunoprecipitation (ChIP) followed by next generation sequencing (ChIP-Seq) is a powerful technique to study transcriptional regulation. However, the requirement of millions of cells to generate results with high signal-to-noise ratio precludes it in the study of small cell populations. Here, we present a tagmentation-assisted fragmentation ChIP (TAF-ChIP) and sequencing method to generate high-quality histone profiles from low cell numbers. The data obtained from the TAF-ChIP approach are amenable to standard tools for ChIP-Seq analysis, owing to its high signal-to-noise ratio. The epigenetic profiles from TAF-ChIP approach showed high agreement with conventional ChIP-Seq datasets, thereby underlining the utility of this approach.

## Introduction

Chromatin immunoprecipitation coupled with next generation sequencing (ChIP-Seq) is a powerful and unbiased approach to study genome-wide DNA–protein interactions and epigenetic modifications (1). However, the prerequisite of huge starting material (millions of cells) limits its utility in studying rare cell types (2). First, sonication, the by far most popular method for fragmentation in ChIP-Seq experiments, can destroy the epitope used for immunoprecipitation, especially when the material is limited (3). The alternative approach of micrococcal nuclease-based digestion (MNase) is hard to control in its efficacy and saturation, and it also shows some degree of sequence-dependent biases (4, 5, 6). Second, the addition of sequencing adaptors for the generation of final libraries involves steps where the limitation of ligation and loss of material during purification steps can result in libraries with low complexity.

Recently, there have been several attempts to adapt ChIP-Seq protocols to address these limitations to apply them to samples with low number of cells (7, 8). One such method, called FARP-ChIP, used nontarget cells for protection during sonication. To prevent the loss of DNA during library preparation, a biotinylated synthetic DNA (biotin-DNA) is used as a carrier DNA. The approach was successfully implemented to obtain the epigenetic profile from samples of 500 mouse embryonic stem cells. However, it required deep sequencing runs (~100 million reads), and the number of reads mapping to the DNA of the target cell type was low (~16%), which makes this method less feasible for many applications and also cost-intensive. Some other recent methods used prior ligation of barcoded adaptors to the chromatin digested by MNase, followed by a computational demultiplexing strategy to obtain profiles from samples of low cell numbers (9). The barcoding strategy was shown to dramatically reduce the number of cells required for each profile and can also remove the biases arising from different chromatin preparations. However, the method still initially requires samples of 10,000–100,000 cells as the starting material. Another approach, microscale μChIP-Seq, was used to generate the profile from samples of 500 cells. However, the method is a scaled-down version of the conventional ChIP-Seq approach with samples subdivided at the level of immunoprecipitation (10). The method ChIPmentation uses Tn5 transposon-mediated tagmentation for preparation of libraries as an alternative to the ligation-based library preparation methods (11). This reduces the hands-on time for library preparation and input requirements. However, this approach uses sonication for fragmenting the chromatin before immunoprecipitation. Moreover, this method still uses a large batch preparation of chromatin and uses subsequent splitting of the sample to generate the profile from samples of 10,000 cells. Recently, the CUT&RUN approach was implemented to generate profiles from samples of 100 cells using antibody-targeted micrococcal nuclease (12). The released and captured DNA was used to generate Illumina-compatible libraries.

Here, we describe an alternative approach for ChIP that uses tagmentation-assisted fragmentation of chromatin (TAF-ChIP) with hyperactive Tn5 transposase from Illumina. The method uses limited sonication power only for nuclear lysis and Tn5 activity for chromatin fragmentation. We have used this approach to generate high-quality

[1]Institute of Developmental Biology and Neurobiology, University of Mainz, Mainz, Germany  [2]Department of Pharmacology, University Medical Center, Johannes Gutenberg University of Mainz, Mainz, Germany  [3]Center for Thrombosis and Hemostasis Mainz, Mainz, Germany  [4]Institute of Medical Biostatistics, Epidemiology and Informatics, Mainz, Germany  [5]Faculty of Biology, Johannes Gutenberg University Mainz, Mainz, Germany

Correspondence: akhtar@uni-mainz.de; bergerc@uni-mainz.de
Apurva Kulkarni present address is Indian Institute of Science Education and Research (IISER) Pune, Maharashtra, India
*Piyush More and Steffen Albrecht contributed equally to this work

datasets from as few as 100 human and 1,000 *Drosophila* cells. This approach has minimal hands-on time and does not involve labor-intensive library preparation workflow. Furthermore, it could be easily implemented to any type of cells. Comparisons of TAF-ChIP results with ENCODE datasets, CUT&RUN, and conventional ChIP-Seq performed in identical cell types demonstrate the utility of this approach. We expect our approach to be especially useful in conditions where the amount of sample is the limiting factor, such as material isolated from animals and clinical samples.

# Results

## Method overview

There are two challenging steps in generating high-quality ChIP-Seq datasets from samples with a very low number of cells. First, the fragmentation of chromatin without compromising the integrity of the associated proteins; second, the generation of Illumina-compatible sequencing libraries, which requires the purified DNA to undergo multiple manipulation steps, namely, end-repair, ligation of the sequencing adaptors, and PCR amplification. These steps also require bead-based purification of nonamplified DNA, where any potential loss of DNA can severely compromise the completion of successful libraries, especially when the starting amount of DNA is low. Furthermore, the intermediate steps can also be the source of variability.

To overcome these limitations, we used tagmentation as a tool to fragment the DNA. Tn5-mediated tagmentation had been previously used for the addition of sequencing adaptors on immunoprecipitated material, when preparing ChIP libraries and genomic DNA libraries.

Here, we instead used Tn5 activity to fragment the intact chromatin during immunoprecipitation. This approach has two major advantages. First, there is no need to fragment the chromatin before immunoprecipitation. Therefore, this strategy prevents potential loss of DNA–protein interactions during fragmentation, especially when compared with sonication. Furthermore, sonication is extremely variable between different machines, even if they are of the same specifications. Second, our tagmentation reactions use the hyperactive Tn5 transposomes that are preloaded with sequencing adaptors (13). Thus, after proteinase K inactivation, the immunoprecipitated material can be directly PCR-amplified. This results in a one-step DNA library generation, which overcomes the limitation in efficiency of ligation and also avoids intermediate purification steps, thereby preventing loss of material (14). After PCR amplification, the amplified libraries are bead-purified.

## Application of the TAF-ChIP approach on sorted *Drosophila* NSCs and human K562 cells

For TAF-ChIP samples, the cells were directly sorted into immunopreciptation buffer owing to the low FACS sheath fluid volume and directly preceded to nuclear lysis with low energy sonication. The low-energy sonication used here did not result in any visible fragmentation of chromatin. The nonfragmented chromatin was subjected to immunoprecipitation and tagmentation. After tagmentation, enzymes

and background regions were washed away with subsequent high-stringency washes. DNA was purified and PCR-amplified to generate Illumina-compatible DNA libraries (see the Materials and Methods section for further details) (Figs 1 and S1A). For conventional ChIP-Seq samples, cells from *Drosophila* larval brain were sorted, pelleted, and resuspended in lysis buffer, as described earlier (15). Upon immunoprecipitation with specific antibodies, the DNA was extracted and converted into DNA libraries (Fig S1B). For the purpose of this study, we used two different types of starting materials: type II neural stem cells (NSCs) from *Drosophila* larval brain and human K562 cells, a human immortalized myelogenous leukemia line. We used formaldehyde to fix freshly dissected *Drosophila* larval brains or harvested K562 cells. The dissected *Drosophila* larval brains expressed a GFP-tagged deadpan (Dpn) protein under the control of its endogenous enhancer, which is a transcription factor only present in NSCs in the brain. This GFP was used to sort NSCs from wild-type larvae, as described earlier (16).

FACS-sorting of wild-type NSCs is not applicable to obtain the ~1 million cells necessary to generate a conventional ChIP-Seq dataset, as one *Drosophila* brain consists of approximately 400 NSCs only. Thus, to compare the TAF-ChIP with the conventional ChIP-Seq protocol, we used the Gal4/UAS binary expression system to express a constitutively active Notch protein (*Notch^intra*) in all type II NSCs (UAS/GAL4 system; *wor*-Gal4; *ase*-Gal80 fly line), also expressing UAS-CD8-GFP (17). The expression of constitutively active *Notch^intra* protein results in a massive over-proliferation of cells with the properties of type II NSCs amenable to cell sorting for conventional ChIP-Seq (18).

We sorted type II NSCs from this line with identical settings as above, for TAF-ChIP (1,000 cells) as well as for conventional ChIP-Seq (1.2 million cells). For obtaining 100 K562 cells, we stained the cells with Hoechst dye and used FACS for collecting samples with the precise number of cells. To benchmark our TAF-ChIP datasets from K562 cells, we used publicly available datasets from the ENCODE project (19, 20).

The Tn5 tagmentation is preferably carried out in the open chromatin region because of higher accessibility (which is the basis of the ATAC-Seq approach), and thus, these regions can get over-represented (21). To distinguish from this scenario and to get a better estimate of background signal, we also performed TAF-ChIP experiments with histone H3.

## Detailed evaluation of TAF-ChIP

To investigate in detail the performance of TAF-ChIP against both the conventional ChIP-Seq and the recently described CUT&RUN low amount method, we used receiver-operating characteristic (ROC) curves and precision-recall (PRC) curves (7). Towards this goal, we compared the peaks in K562 cells for the TAF-ChIP datasets, conventional ENCODE datasets, and CUT&RUN datasets for 100, 3,000, and 6,000 cells at various false discovery rate (FDR) cutoffs and using the *replicated peaks* of the conventional ENCODE dataset as reference (12, 19). K652 curves were calculated by mapping peaks to 5 kb non-overlapping genomics windows. Similarly, we also compared peaks for TAF-ChIP and for conventional ChIP-Seq datasets from *Drosophila* UAS-derived NSCs at various FDR cutoffs and using the first replicate of the conventional ChIP-Seq

1. Sort fixed cells directly in the lysis/IP buffer

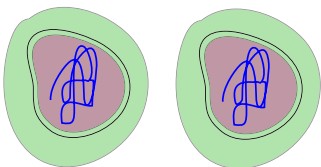

2. Nuclear lysis with mild sonication



3. Couple antibodies to beads with blocking

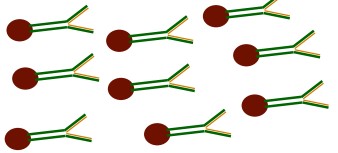

4. Incubation of Ab-beads with chromatin

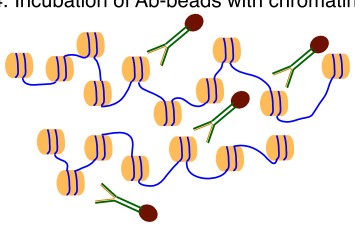

5. Mild wash followed by tagmentation

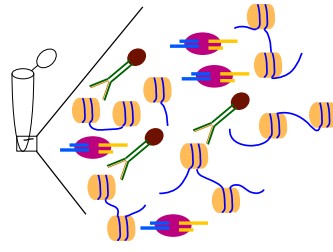

Tagmentation assisted fragmentation

6. Higher stringency wash to remove background

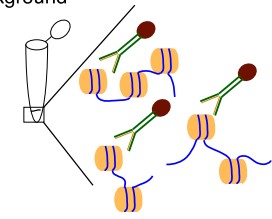

7. Reverse cross linking, Prot. K inactivation and PCR

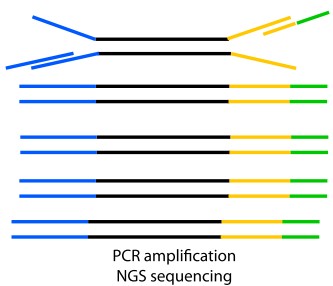

PCR amplification
NGS sequencing

**Figure 1. Schematic overview of TAF–ChIP approach.**
**(1)** Formaldehyde fixed cells were directly sorted into radio immunoprecipitation (RIPA) buffer (see the Materials and Methods section for details). **(2)** The cells were briefly sonicated at low intensity to break open the nuclei. **(3)** Antibodies were coupled to magnetic beads in the presence of blocking reagents. **(4)** Antibody-coupled beads were added to the cell lysate and incubated overnight at 4°C. **(5)** The tagmentation reaction was performed after initial washes with low salt IP buffer and homemade tagmentation buffer. **(6)** The tagmentation reaction and the background regions (not anchored by antibody interaction) were washed away with subsequent high-stringency washes. **(7)** The proteinase K was heat-inactivated and the material was PCR-amplified without purification.

dataset as reference. NSC curves were calculated by mapping peaks to 1-kb non-overlapping genomic windows. The peaks were always obtained with MACS2 peak calling algorithm using either input (conventional ChIP-Seq) or H3 datasets (TAF-ChIP) as controls.

For human K562 cells, both the ROC curves and the precision-recall curves showed that the 100-cell TAF-ChIP dataset was comparable with the reference ENCODE replicate, as well as to 3,000 and 6,000 cells CUT&RUN datasets, outperforming the 100-cell CUT&RUN dataset (Fig 2A and B). Only ~500 peaks were called at 5% FDR for the 100-cell CUT&RUN dataset. This could be due to high occurrence of noise in the 100-cell CUT&RUN dataset, which can be observed in the genome browser profile (Fig S1C). The CUT&RUN method on 100 cells was not able to recall more than 75% of the reference even though the peak calling parameters had no restrictions.

For *Drosophila* NSCs, the ROC and PRC curves showed that our TAF-ChIP approach has a comparable performance to the inter-replicate results of conventional ChIP-Seq (Fig 2C–F).

We next compared the datasets by hierarchical clustering using a similarity measure based on the Jaccard index calculated on sets of genomic windows from peaks defined at 5% FDR. The conventional H3K4Me3 ChIP-Seq datasets from *Drosophila* NSCs (tumor derived) clustered together with H3K4Me3 TAF-ChIP datasets from tumor NSCs rather than with wild-type NSCs (Fig 2G). For K562 cells, the H3K27Me3 TAF-ChIP datasets clustered together with the corresponding

ENCODE dataset and with CUT&RUN datasets from higher cell numbers (Fig 2H). Consistent with our ROC curve and PRC curve analysis, the 100-cell CUT&RUN dataset showed lower similarity to the rest of the datasets.

We also plotted the hierarchical clustering for H3K9Me3 and H3K27Me3 with other histone ChIP-Seq datasets included as control. The TAF-ChIP datasets always clustered together with their corresponding ENCODE datasets rather than with unrelated histone ChIP-Seq (Fig S1D and E). The TAF-ChIP dataset for H3K9Me3 from *Drosophila* NSCs (tumor) also clustered together with conventional ChIP-Seq performed in identical NSCs (Fig S1F).

### Comparison of 100-cell TAF-ChIP with ENCODE dataset

To further test the applicability of TAF-ChIP, we next used corresponding conventional ChIP-Seq datasets from the ENCODE project for benchmarking.

The H3K27Me3 TAF-ChIP and H3K9Me3 TAF-ChIP from samples of 100 cells showed similar profiles when compared with the corresponding ENCODE datasets, as visualized through genome browser tracks (Fig 3A and B), and also had good agreement between the replicates when Pearson's correlation coefficient was calculated using average signal in each 2-kb non-overlapping genomic window (Fig S2A and B). The metagene profile for H3K27Me3 and H3K9Me3

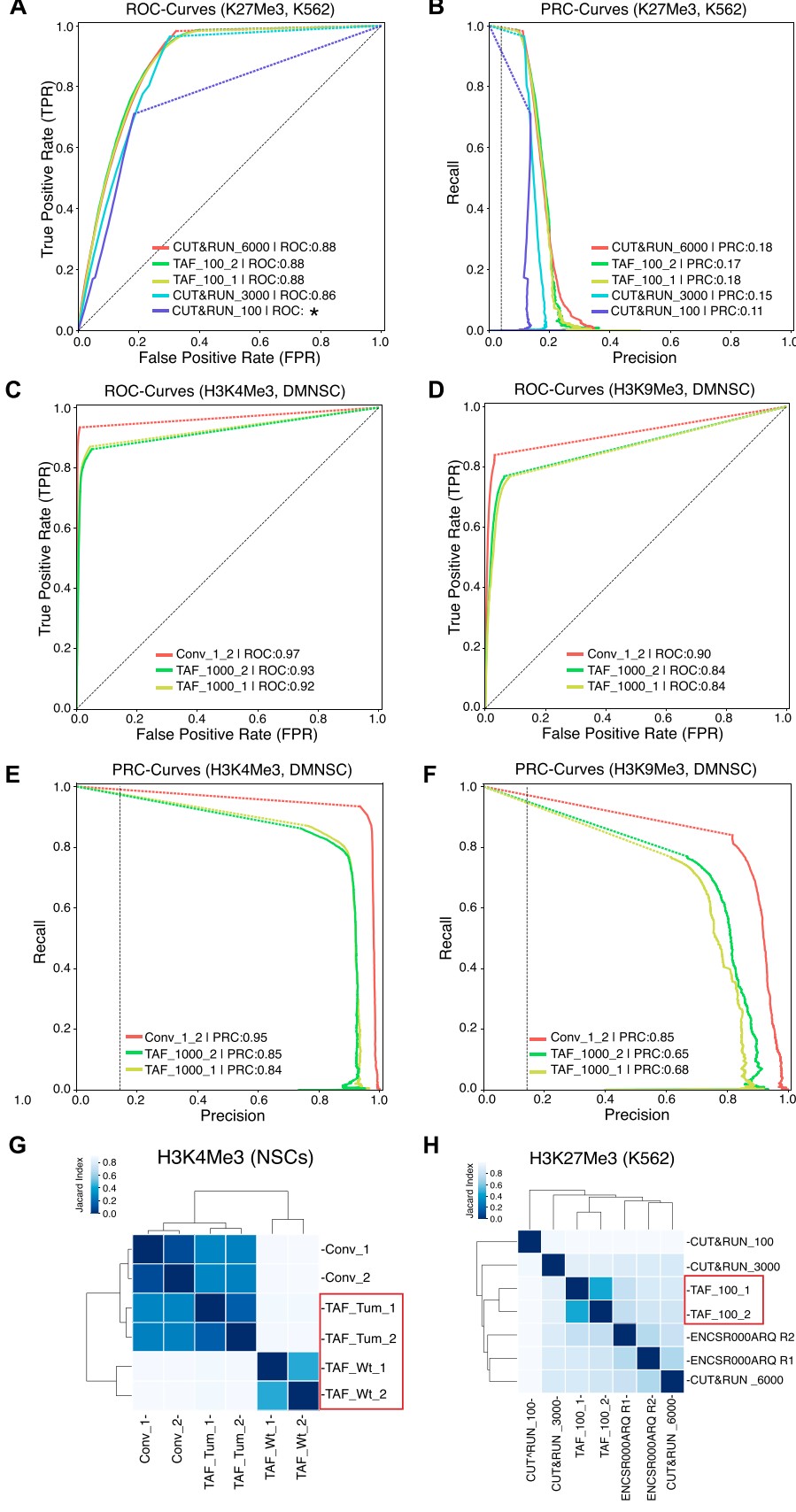

**Figure 2. Comparison of TAF–ChIP with conventional ChIP-Seq and with the CUT&RUN low amount method.**
**(A)** ROC curves of TAF-ChIP and CUT&RUN for H3K27Me3 in K562 cells. The ROC curves were plotted using as reference replicated peaks of the conventional ChIP-Seq ENCODE dataset selected at 5% FDR cutoff (downloaded from the ENCODE database). No FDR cutoff was used to define peaks for TAF-ChIP replicates and the CUT&RUN datasets with MACS2. Peaks were mapped to 5 kb non-overlapping genomic windows to calculate true-positive rate or recall, false-positive rate and precision for a changing P-Value threshold. Area under the curve (AUC) is indicated in the legend in decreasing order, and the * indicates the failure to faithfully calculate the AUC. **(B)** Precision-recall curve for TAF-ChIP and CUT&RUN datasets for H3K27Me3 in K562 cells. **(C, D)** ROC curves of TAF-ChIP and conventional ChIP-Seq in *Drosophila* NSCs. The ROC curves for H3K4Me3 (C) and H3K9me3 (D) were plotted using as reference peaks of the first conventional ChIP-Seq replicate selected at 5% FDR cutoff. No FDR cutoff was used to define peaks for TAF-ChIP replicates and the second conventional ChIP-Seq replicate. Peaks were mapped to 1 kb non-overlapping genomic windows to calculate true-positive rate or recall, false-positive rate, and precision. AUC is indicated in the legend in decreasing order. **(E, F)** Precision-recall curve for TAF-ChIP and conventional ChIP-Seq in *Drosophila* NSCs. Using same references and data as above, precision-recall curves were plotted for H3K4Me3 (E) and H3k9Me3 (F). **(G)** Comparison of the genomic window sets for *Drosophila* brain-derived wt NSCs analyzed for H3K4Me3 binding by TAF-ChIP (TAF_Wt), and *Drosophila* tumor-derived NSCs analyzed by TAF-ChIP or conventional ChIP-Seq (TAF_Tum and Conv). The TAF-ChIP samples are highlighted by a red rectangular box. The heat map indicates pairwise similarity according to the Jaccard index. Axes show results of hierarchical clustering. **(H)** The Jaccard index and hierarchical clustering, as described in (G), to compare H3K27Me3 binding in K562 cells. The comparison was performed for 100 cells TAF-ChIP samples (highlighted with a red rectangular box), to CUT&RUN method with 100, 3,000, and 6,000-cell samples, and to conventional ChIP-Seq (ENCODE) (12, 19).

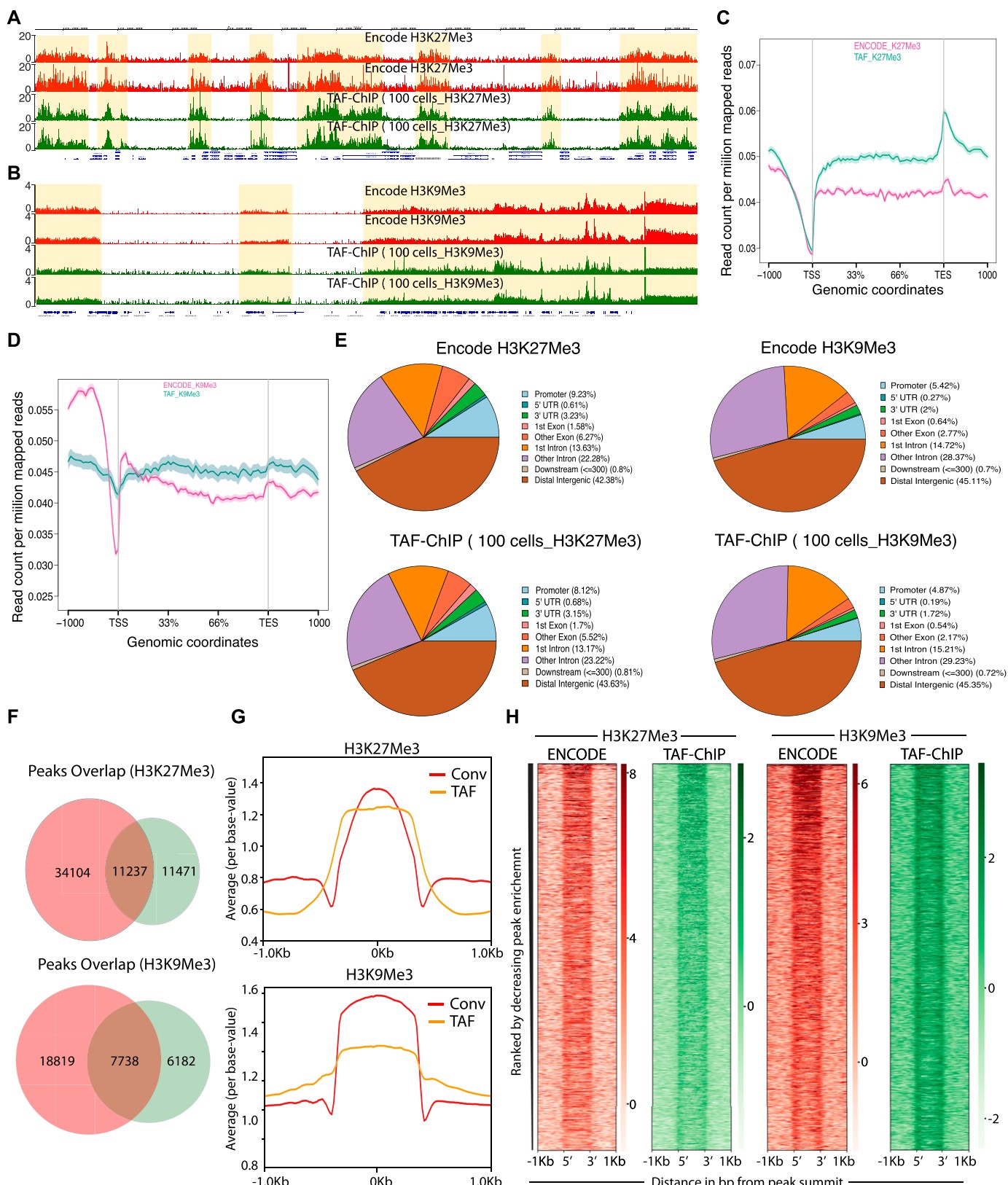

**Figure 3. TAF-ChIP results from 100 K562 cells are comparable with conventional Encode ChIP-Seq datasets.**
**(A, B)** Genome browser track example of H3K27Me3 and H3K9Me3 (A and B, respectively) ChIP performed in 100 FACS sorted K562 cells with TAF-ChIP approach and corresponding K562 conventional ChIP-Seq datasets from the ENCODE project in duplicates, as indicated in the labels. The label below the tracks shows the gene model

showed decrease at the transcription start sites (TSSs) and higher signal on the gene body, similar to the profile obtained with the ENCODE dataset (Fig 3C and D). We used the MACS2 peaks calling algorithm for identifying the peaks in both TAF-ChIP and ENCODE datasets, with identical parameters. The corresponding input samples, fragmented input control for ENCODE and H3 TAF for TAF-ChIP, were used as controls for peak calling. The annotation of peaks identified in the TAF-ChIP dataset and in the corresponding one from ENCODE showed similarity in distribution of overlapping genomic features, for both H3K27Me3 and H3K9Me3 datasets (Fig 3E). The overlap between the peaks called for ENCODE and TAF-ChIP was 50% and 56% for H3K27Me3 and H3K9Me3, respectively (Fig 3F). Next, we divided the peaks into 10 different quantiles according to FDR, with quantile 1 associated with the lowest FDR peaks and quantile 10 associated with the highest. The FDR quantile recovery analysis for H3K27Me3 and H3K9Me3 peaks compared with replicated peaks of ENCODE was higher for lower FDR quantiles, at around 60 and 70%, respectively (Fig S2C and D). The fraction of reads in peaks called with TAF-ChIP had also similar distribution profile when compared with the ENCODE ChIP-Seq. However, the level of this enrichment was smaller for TAF-ChIP (Fig 3G). Nonetheless, the heat maps generated for all the peaks identified in ENCODE ChIP-Seq datasets and sorted according to the intensity in the ENCODE ChIP-Seq, showed profiles similar and comparable with TAF-ChIP datasets from 100 K562 cells (Fig 3H).

### TAF-ChIP performed on *Drosophila* NSCs shows high agreement with conventional ChIP-Seq

To compare TAF-ChIP with conventional ChIP-Seq, we analyzed both H3K4Me3 and H3K9Me3 histone marks, from identical cell types, as described above. The TAF-ChIP generated datasets showed similar signal-to-noise ratio when compared with corresponding conventional ChIP-Seq datasets, as visualized through genome browser tracks (Fig 4A and B). The TAF-ChIP data also showed high degree of mappability and low level of sequence duplication. The uniquely mapped reads for H3K4Me3 samples were at ~80%. The unique mapping rate for H3K9Me3 was lower at ~60%, yet this can be expected because of prevalence of this mark at repeat elements and transposons (Fig S3A). The replicates also showed good concordance between themselves when Pearson's correlation coefficient was calculated using average signal in each 500-bp non-overlapping genomic windows (Fig S3B and C). The metagene profile for H3K4Me3 normalized to H3 and IgG control showed higher signal at the TSSs, consistent with the higher enrichment of this mark at the promoters (Fig 4C). On the other hand, the metagene profile for H3K9Me3 showed higher enrichment over gene body (data not shown). Furthermore, the qPCR analysis of TAF-ChIP and

conventional ChIP showed comparable enrichment for enriched loci and similar level of background at nonenriched locus (Fig S3D). Next, we used the MACS2 software (21) to identify peaks in the TAF-ChIP and in conventional ChIP-Seq datasets. The fragmented input control and H3 TAF-ChIP datasets were used as input control for conventional ChIP-Seq and TAF-ChIP datasets for peak calling, respectively. The deposition of H3K9Me3 was mostly on intergenic regions; therefore, we used peak coordinates to generate the normalized metagene profile (Fig 4D). The annotation of peaks obtained from TAF-ChIP and ChIP-Seq showed a higher degree of similarity for the H3K4Me3 mark than for the H3K9Me3 mark, the latter displaying more overlap to promoters and less to intergenic regions in conventional ChIP-Seq (Fig 4E). Nevertheless, consistent with the expectation, the large fraction of H3K4Me3 peaks was at the promoters, whereas the most H3K9Me3 peaks were at the distal intergenic regions. Next, we calculated the overlap between the peaks called for conventional ChIP-Seq and TAF-ChIP datasets using the ChIPpeakAnno package from Bioconductor (22). The peaks called for H3K4Me3 showed 85% overlap between the conventional and TAF-ChIP approaches at 5% FDR. The peaks called at 5% FDR for H3K9Me3 had 68% of overlap between the conventional and TAF-ChIP approach (Fig 4F).

Next, we performed the peak recovery in different FDR quantiles, as explained before for K562 datasets. Using one H3K4me3 conventional ChIP-Seq replicate as reference, TAF-ChIP recalled ~99% of the peaks until quantile 6, and was comparable with the other replicate of the conventional ChIP-Seq (Fig S3E). The relationship between recall and FDR was very weak for H3K9Me3; however, it was still similar to conventional ChIP-Seq (Fig S3F). The read distribution at the peaks still showed enrichment for TAF-ChIP, albeit to a lower level when compared with conventional ChIP-Seq datasets (Fig 4G). The analysis for saturation of peak recall showed higher recall of peaks for H3K4Me3 at shallow sequencing depth, whereas for H3K9Me3, the number of recalled peaks continued to increase with increasing sequencing depths (Fig S3G and H). This was consistent with the observed tendencies for point-source histone modifications (such as H3K4Me3) and histone modifications with broad domains of enrichments (such as H3K9Me3) (20). The distributions of reads at genomic locations generated for TAF-ChIP and conventional ChIP-Seq datasets and sorted according to the intensity in the conventional ChIP-Seq resulted in similar and comparable profiles (Fig 4H).

### TAF-ChIP gave consistent results with variable numbers of cells used as starting material

After establishing TAF-ChIP using low number of cells and its subsequent benchmarking against conventional ChIP-Seq performed

and the y-axis represents normalized read density in reads per million. The enriched regions are highlighted with shaded box. **(C, D)** Metagene profiles of H3K27Me3 and H3K9Me3 (C and D, respectively) with standard error to the mean for all the genes, −1,000 bp upstream of TSS and +1,000 bp downstream of transcription end sites (TES). Read counts per million of mapped reads is shown on the y-axis, whereas the x-axis depicts genomic coordinates. **(E)** Genomic distribution of annotated peaks obtained from the ENCODE datasets and TAF-ChIP (100 K562 cells), for indicated histone marks. Note the majority of H3K27Me3 and H3K9Me3 peaks are at the intergenic regions, consistent with the expectation. **(F)** Overlap between the peaks identified from the ENCODE and TAF-ChIP datasets, for the indicated histone modifications (see the Materials and Methods section for further details). **(G)** Average profile of TAF-ChIP and corresponding ENCODE ChIP-Seq centered at the peaks for the indicated histone modifications. The y-axis depicts average per base value into the peaks, whereas x-axis depicts genomic coordinates centered at the peaks. **(H)** Distributions of reads at gene locations of indicated histone modifications from ENCODE ChIP-Seq and TAF-ChIP method, centered at the peaks (−1 kb to +1 kb). Rows indicate all the peaks and are sorted by decreasing affinities in the ENCODE ChIP-Seq datasets. The color labels to the right indicate the level of enrichment.

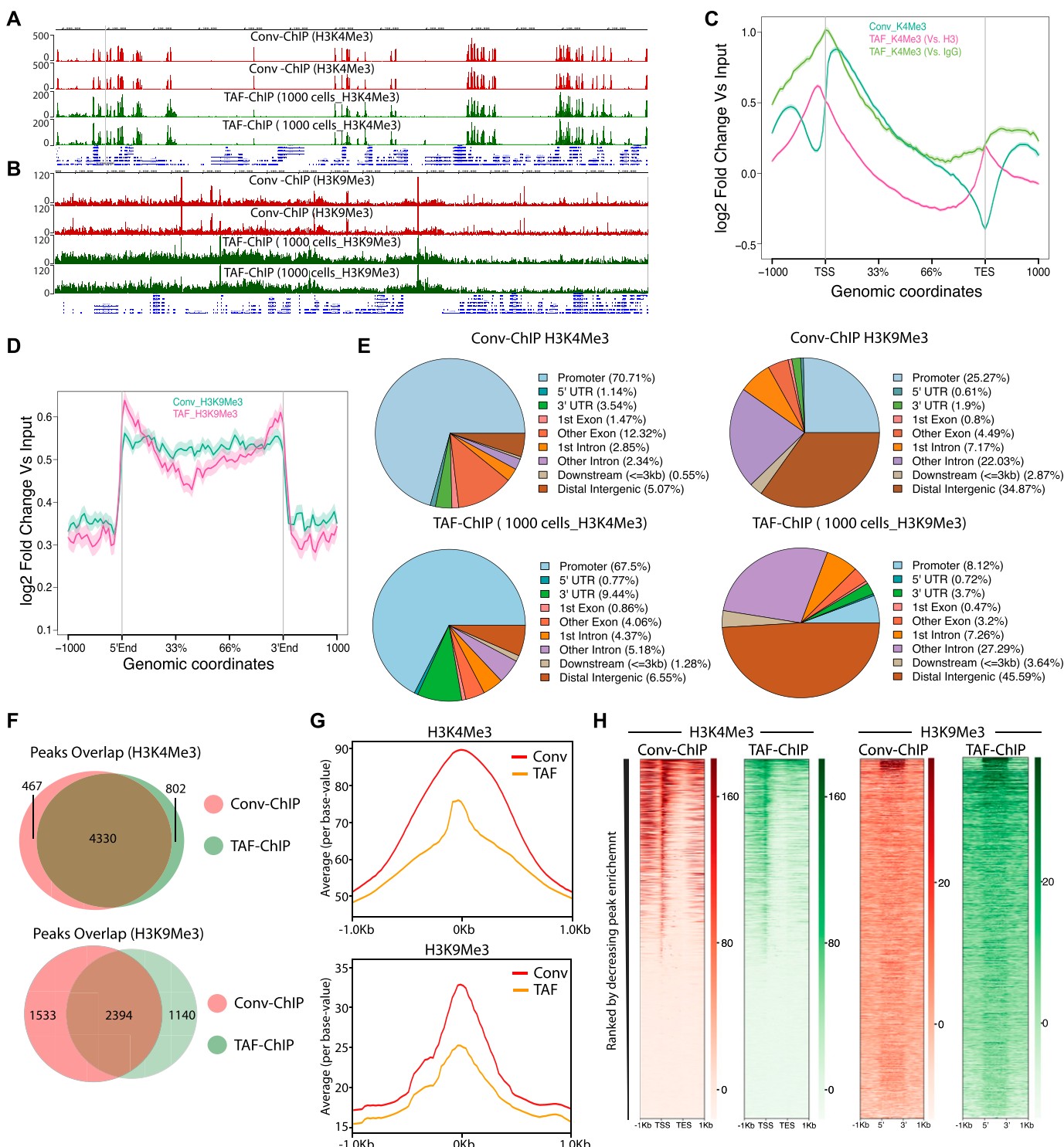

**Figure 4. TAF-ChIP results from low number of NSCs are comparable with conventional ChIP-Seq (Conv-ChIP).**
**(A, B)** Genome browser track example of H3K4Me3 and H3K9Me3 ChIP (panel A and B, respectively) performed in FACS-sorted NSCs with conventional ChIP-Seq (1.2 million cells) and TAF-ChIP (1,000 cells), as indicated by the labels. The label below the tracks shows the gene model and the y-axis represents normalized read density in reads per million (rpm). **(C)** Metagene profiles of H3K4Me3 with standard error to the mean for all genes, −1,000 bp upstream of TSS and +1,000 bp downstream of transcription end sites (TESs). Log2-fold changes against input controls are shown on the y-axis, whereas the x-axis depicts genomic coordinates. **(D)** Metagene profiles of H3K9Me3 with standard error to the mean for enriched regions, −1,000 bp upstream and +1,000 bp downstream of peaks. Log2-fold changes against input control are shown on the y-axis, whereas the x-axis depicts genomic coordinates. **(E)** Distribution of annotated peaks obtained from conventional ChIP-Seq and TAF-ChIP, for indicated histone marks. Note that most H3K4Me3 and H3K9Me3 peaks are at the promoters and at the intergenic regions, respectively, consistent with the expectation. **(F)** Overlap between the peaks identified from conventional ChIP-Seq and TAF-ChIP datasets, for the indicated histone modifications. MACS2 software with identical

in identical cells, we next assayed whether TAF-ChIP can give comparable results with similar resolution, when variable numbers of cells are used as the starting material. Towards this goal, we resorted to use wild-type NSCs from *Drosophila* brains. We sorted two samples containing 1,000 and 5,000 NSCs, respectively. The TAF-ChIP generated datasets from 1,000 and 5,000 NSCs resulting in nearly identical profiles, as visualized through genome browser tracks (Fig S4A). The distributions of reads at genomic locations generated from 1,000 NSCs and 5,000 NSCs also showed comparable profiles (Fig S4B). The read distribution in the peaks for samples with 1,000 NSCs and 5,000 NSCs were also comparable with each other (Fig S4C). Altogether, these results suggest that TAF-ChIP is amenable to conditions when starting material is variable to few folds and would produce similar results.

# Discussion

Here, we present an easy, TAF-ChIP and sequencing method to generate high-quality datasets from samples with low cell numbers. The workflow of TAF-ChIP contains fewer steps than conventional ChIP-Seq with minimum hands-on time during library preparation, preventing loss of material and potential user introduced variability. Because of tagmentation during immunoprecipitation, the cells can be directly sorted into the IP/lysis buffer. This eliminates the centrifugation step to collect the cells, which can also lead to potential loss of material. Also, unlike ATAC-Seq where intact nuclei are tagmented and partial tagmentation is used to study chromatin accessibility, our approach tagments after nuclear lysis (21). The metagene profile of H3 TAF-ChIP dataset did not show any enrichment for TSS, suggesting our method resulted in tagmentation without any visible biases for open chromatin regions (Fig S4D and E). Furthermore, we also showed the application of TAF-ChIP for both open chromatin marks such as H3K4Me3 as well as for repressive marks such as H3K9Me3 and H3K27Me3. TAF-ChIP is easier to implement than MNase-based approaches that leads to overdigestion of chromatin, and results in one-step generation of Illumina-compatible libraries. The TAF-ChIP approach is suitable for assaying factors for which the chromatin association might be dependent on RNA, as Tn5 does not perturb the RNA intermediate. Also, the tagmentation does not show any sequence-dependent biases, in contrast to other restriction-based protocols (6, 13). Furthermore, the approach does not need any specialized equipment and, thus, can be implemented in a standard molecular biology laboratory.

We have used here the Tn5 transposase from Nextera XT DNA library kit; however, TAF-ChIP could be easily implemented with Tn5 loaded with different unique molecular indices (13). This could be easily implemented in massively parallel TAF-ChIP-Seq applications and may even further decrease the required starting material.

Moreover, as this approach can be used for various cell types, it could be also combined with a nontarget cell type used as "spike-in" and DNA carrier.

We showed that TAF-ChIP datasets have signal-to-noise ratios that are comparable with conventional ChIP-Seq datasets and, thus, are amenable to standard bioinformatics pipelines for ChIP-Seq analysis. Our evaluation of TAF-ChIP datasets showed results comparable with conventional ChIP-Seq and better than CUT&RUN, a comparable low amount method. For histone marks, we demonstrated the use of H3 TAF-ChIP as an input control for better background estimation. However, in some conditions and perturbation experiments, the distribution of H3 might have weak to strong biases. An alternative control for TAF-ChIP could be also immunoprecipitation with IgG from similar species (Fig 4C). Although the genome browser profiles obtained from a sample of 100 K562 cells showed slightly inferior signal-to-noise ratio compared with the conventional datasets from the ENCODE project (Fig 2A), the peaks identified were mostly overlapping, especially at lower FDRs. The peaks that were unique to either TAF-ChIP or to the conventional method still showed higher read coverage compared with randomly selected regions of comparable size (Fig S5A–D). This suggests that thresholding (based on FDR) implemented by the peak caller software might have hindered their identification in one of the datasets. Furthermore, we suspect that the signal-to-noise ratio can be improved by pooling the samples tagmented with different indices before washes, and using the demultiplexing strategy to obtain the data.

We have also been able to generate the profile of a RNA-modifying enzyme, recently shown in vertebrates to associate with chromatin, by performing TAF-ChIP using 1,000 cells from transgenically tagged *Drosophila* and antibody directed against the tag (unpublished results). We conclude that the only limiting factors defining the low cell number sample providing biologically meaningful TAF-ChIP results are the availability of a good antibody and a reasonable number of binding sites in the genome. We have shown that TAF-ChIP provides reliable datasets from samples of as low as 100 isolated cells without requiring prior isolation of nuclei and with an extremely easy and straightforward workflow; therefore, we expect that TAF-ChIP will be very useful when access to higher numbers of cells is limited.

# Materials and Methods

### Antibodies

The following antibodies were used in this study. For H3K4Me3 ChIP, antibody from Abcam (Cat. no. ab8580) was used. H3K9Me3 ChIP was performed using an antibody from Active Motif (Cat. no. 39161), H3K27Me3 with Active Motif (Cat. no. 39155), and H3 (ab1791; Abcam).

parameters (see the Materials and Methods section for details) was used to identify the peaks against the respective input controls, and those present in both replicates were considered for the comparison. **(G)** Average profile of TAF-ChIP and conventional ChIP-Seq centered at the peaks for the indicated histone modifications. The y-axis depicts average per base value into the peaks, whereas the x-axis depicts genomic coordinates centered at the peaks. **(H)** Distributions of reads at gene locations of indicated histone modifications from conventional ChIP-Seq and TAF-ChIP. Rows indicate all the peaks and are sorted by decreasing affinities in the conventional ChIP-Seq datasets. The color labels to the right indicate the level of enrichment.

## Fixation and cell sorting from *Drosophila* larval brain

Briefly, required number of larval brains after 48 h of larval hatching were dissected in PBS. After dissection, larval brains were fixed with 1% formaldehyde in PBS for 10 min at room temperature, followed by quenching of the fix with 125 mM glycine. The larval brains were dissociated and resuspended according to the previously established method (16). The cells were then sorted on BD FACSAria according to the strength of GFP and size of the NSCs, resulting in a pure population of type II NSCs.

## Fixation and cell sorting of K562 cells

K562 cells cultured in RPMI medium (supplemented with 10% Fetal Bovine Serum), at 37°C and 5% $CO_2$, were fixed for 10 min at room temperature with 1% formaldehyde. The crosslink was quenched with 125 mM glycine, and sorted on BD FACSAria cell sorter using Hoechst stain. A total of 100 K562 cells were directly sorted in RIPA 140 mM (10 mM Tris-Cl, pH 8.0, 140 mM NaCl, 0.5 mM EDTA, pH 8.0, 1% Triton X-100, and 0.1% SDS).

## Conventional ChIP-Seq and library preparation

Fixed cells (1.2 million FACS sorted NSCs per replicate) were resuspended in 140 mM RIPA (10 mM Tris-Cl pH 8.0, 140 mM NaCl, 0.1 mM EDTA pH 8.0, 1% Triton X-100, and 0.1% SDS) and subjected to 14 cycles of sonication on a bioruptor (Diagnode), with 30 secs "ON"/ "OFF" at high settings. After sonication, the samples were centrifuged at 14,000 *g* for 10 min at 4°C and the supernatant was transferred to a fresh tube. The extracts were incubated overnight with 2 µg of specific antibody at 4°C with head-over-tail rotations. After overnight incubations, 20 µl of blocked Protein A and G Dynabeads were added to the tubes and further incubated for 3 h to capture the antibodies. The beads were separated with a magnetic rack and were washed as following: once with 140 mM RIPA (10 mM Tris-Cl, pH 8.0, 140 mM NaCl, 0.1 mM EDTA, pH 8.0, 1% Triton X-100, and 0.1% SDS), four times with 250 mM RIPA (10 mM Tris-Cl, pH 8.0, 250 mM NaCl, 0.1 mM EDTA, pH 8.0, 1% Triton X-100, and 0.1% SDS) and twice with TE buffer, pH 8.0 (10 mM Tris-Cl, pH 8.0 and 0.1 mM EDTA pH 8.0). After the immunoprecipitation, samples were RNase-treated (NEB) and subjected to Proteinase K treatment for reversal of cross-links, 12 h at 37°C and at least 6 h at 65°C. The samples after proteinase K treatment were subjected to phenol–chloroform extraction. After precipitating and pelleting, the DNA was dissolved in 30 µl of TE buffer, pH 8.0. The recovered DNA was converted into libraries using NebNext Ultra II DNA library preparation kit, following the manufacturer's protocol.

## TAF-ChIP and library preparation

Cells were fixed for 10 min at room temperature with 1% formaldehyde in PBS. The formaldehyde was quenched with 125 mM glycine, for 5 min at room temperature. The cells were washed once with ice-cold PBS and directly sorted in 240 µl of 140 mM RIPA (10 mM Tris-Cl, pH 8.0, 140 mM NaCl, 0.1 mM EDTA, pH 8.0, 1% Triton X-100, and 0.1% SDS) and sonicated with three cycles at low power settings for breaking the nuclei. Sorting small number of cells (100 cells normally elute in <2 µl) will have no effect on buffer composition owing to small volume. In the meantime, 15 µl of Protein A and G Dynabeads were coupled to 1 µg of specific antibody in the blocking buffer (RIPA 140 mM supplemented with 0.2 mg/ml BSA, 0.05 mg/ml of glycogen, and 0.2 mg/ml of yeast tRNA) for 2-3 h at 4°C. The partially fragmented chromatin was centrifuged at 2,000 *g* for 10 min at 4°C, and the supernatant was transferred to the tube with blocked and antibody-coupled beads. The centrifugation step is optional and the samples can be also directly added to the coupled beads. The samples were incubated at 4°C overnight with head-over-tail rotations. The samples were then washed twice briefly with 300 µl of homemade tagmentation buffer (20 mM Tris(hydroxymethyl) aminomethane, pH 7.6; 10 mM $MgCl_2$; and 20% [vol/vol] dimethylformamide) using magnetic rack for beads separation. The washed beads were resuspended in 20 µl of 1× tagmentation DNA buffer (Nextera XT Kit) containing 1 µl of Nextera DNA tagmentation enzyme and incubated at 37°C for 40 min with constant shaking in a thermoblock at 500 rpm (Eppendorf thermomixer compact). After the tagmentation, the beads were washed as following: once with 140 mM RIPA (10 mM Tris-Cl, pH 8.0, 140 mM NaCl, 0.1 mM EDTA, pH 8.0, 1% Triton X-100, and 0.1% SDS), four times with 250 mM RIPA (10 mM Tris-Cl, pH 8.0, 250 mM NaCl, 0.1 mM EDTA, pH 8.0, 1% Triton X-100, and 0.1% SDS), and twice with TE buffer, pH 8.0 (10 mM Tris-Cl, pH 8.0 and 0.1 mM EDTA, pH 8.0). The samples were subjected to proteinase K treatment in 50 µl of TE buffer, pH 8.0, with 5 µl of 20 mg/ml of proteinase K for overnight at 60°C in thermoblock with shaking at 500 rpm (Eppendorf thermomixer compact). The samples can be then either phenol–chloroform extracted (A) or can be directly PCR-amplified after deactivating proteinase K (B). Phenol–chloroform extraction (A): The volume of the aqueous phase was brought to 100–200 µl by adding TE buffer, and 300 µl of phenol:chloroform (pH 7.7–8.3) was added to the tube. After vortexing, the tubes were spun at 20,000 *g* for 5 min at room temperature, and the aqueous phase was transferred to a fresh tube. The DNA was precipitated by adding 1/10[th] volume of 3 M sodium acetate (pH 5.2), 5 µl of glycogen (20 mg/ml), and 700 µl of 100% ethanol. The precipitation mix was incubated overnight at −80°C followed by centrifugation at 18,800 *g* for 30 min at 4°C. The pellet was washed once with 75% ethanol and resuspended in 30 µl of TE buffer. The DNA was amplified in 100 µl reaction with 1X NEBNext High-Fidelity PCR Mix with 0.4 µM of primers containing molecular indices (listed in Table 1) with the following program: 72°C for 3 min (98°C for 10 s, 63°C for 30 s, and 72°C for 30 s) for 12 cycles, 72°C for 5 min, and hold at 4°C. Deactivating proteinase K (B): the proteinase K was heat-inactivated for 95°C for 5 min and directly amplified in a 100-µl reaction with 1× NEBNext High-Fidelity PCR Mix with 0.4 µM primers containing molecular indices (listed in Table 1) with the following program: 72°C for 3 min {98°C for 10 s, 63°C for 30 s, and 72°C for 30 s} for 12 cycles, 72°C for 5 min, and hold at 4°C.

The PCR reaction was purified with bead-based size selection to remove fragments larger than 1,000 bp. Ampure Xp beads were added to the PCR reaction in a ratio of 0.2× ratio to bind larger fragments. The magnetic beads were separated with the help of magnetic rack and the supernatant was transferred to a fresh tube. Ampure Xp beads were added to the PCR reaction in a ratio of 0.8× to bind the target library. After PCR purification, the libraries were analyzed on Agilent Bioanalyzer for size distribution and the concentration was measured using a Qubit fluorometer. The finished

**Table 1.  Primers used in qPCR and TAF-ChIP library preparation.**

| | |
|---|---|
| Mapk distal Fwd | ATCGGGACCTTAAGCCAAGT |
| Mapk distal Rev | AAACGCTTTTACTGCTGATGG |
| Dock Fwd | GCTCCGGCAAAATCATTAAA |
| Dock Rev | CGCGATTGAAAAACACACAA |
| Dab Fwd | CCCCACAACGCCTTAAAGTA |
| Dab Rev | TTTGCGTCTTCCGTCTCTTT |
| Rca1 Fwd | GGTCACACTGATCCGTACCC |
| Rca1 Rev | CTCCAACTCGAAGGATGACC |
| Primer-Neg Fwd | CCATTAATCGAGGGCTGAAA |
| Primer-Neg Rev | TTGGGGCATAAACAGAGGAC |
| fw ATAC-seq primer, general, no index | AATGATACGGCGACCACCGAGATCTACACTCGTCGGCAGCGTCAGATGT*G |
| rev ATAC-seq primer, Truseq, index 34 CATGGC | CAAGCAGAAGACGGCATACGAGATGCCATGGTCTCGTGGGCTCGGAGATG*T |
| rev ATAC-seq primer, Truseq index, 48 TCGGCA | CAAGCAGAAGACGGCATACGAGATTGCCGAGTCTCGTGGGCTCGGAGATG*T |
| rev ATAC-seq primer, Truseq index, 22 CGTACG | CAAGCAGAAGACGGCATACGAGATCGTACGGTCTCGTGGGCTCGGAGATG*T |
| rev ATAC-seq primer, Truseq index, 15 ATGTCA | CAAGCAGAAGACGGCATACGAGATTGACATGTCTCGTGGGCTCGGAGATG*T |
| rev ATAC-seq primer, Truseq index, 46 TCCCGA | CAAGCAGAAGACGGCATACGAGATTCGGGAGTCTCGTGGGCTCGGAGATG*T |
| rev ATAC-seq primer, Truseq index, 45 TCATTC | CAAGCAGAAGACGGCATACGAGATGAATGAGTCTCGTGGGCTCGGAGATG*T |
| rev ATAC-seq primer, Truseq index, 40 CTCAGA | CAAGCAGAAGACGGCATACGAGATTCTGAGGTCTCGTGGGCTCGGAGATG*T |
| rev ATAC-seq primer, Truseq index, 39 CTATAC | CAAGCAGAAGACGGCATACGAGATGTATAGGTCTCGTGGGCTCGGAGATG*T |
| rev ATAC-seq primer, Truseq index, 38 CTAGCT | CAAGCAGAAGACGGCATACGAGATAGCTAGGTCTCGTGGGCTCGGAGATG*T |
| rev ATAC-seq primer, Truseq index, 37 CGGAAT | CAAGCAGAAGACGGCATACGAGATATTCCGGTCTCGTGGGCTCGGAGATG*T |
| rev ATAC-seq primer, Truseq index, 36 CCAACA | CAAGCAGAAGACGGCATACGAGATTGTTGGGTCTCGTGGGCTCGGAGATG*T |
| rev ATAC-seq primer, Truseq index, 35 CATTTT | CAAGCAGAAGACGGCATACGAGATAAAATGGTCTCGTGGGCTCGGAGATG*T |
| rev ATAC-seq primer, Truseq index, 25 ACTGAT | CAAGCAGAAGACGGCATACGAGATATCAGTGTCTCGTGGGCTCGGAGATG*T |
| rev ATAC-seq primer, Truseq index, 26 ATGAGC | CAAGCAGAAGACGGCATACGAGATGCTCATGTCTCGTGGGCTCGGAGATG*T |
| rev ATAC-seq primer, Truseq index, 27 ATTCCT | CAAGCAGAAGACGGCATACGAGATAGGAATGTCTCGTGGGCTCGGAGATG*T |
| rev ATAC-seq primer, Truseq index, 28 CAAAAG | CAAGCAGAAGACGGCATACGAGATCTTTTGGTCTCGTGGGCTCGGAGATG*T |
| rev ATAC-seq primer, Truseq index, 23 GAGTGG | CAAGCAGAAGACGGCATACGAGATCCACTCGTCTCGTGGGCTCGGAGATG*T |
| rev ATAC-seq primer, Truseq index, 24 GGTAGC | CAAGCAGAAGACGGCATACGAGATGCTACCGTCTCGTGGGCTCGGAGATG*T |
| rev ATAC-seq primer, Truseq index, 42 TAATCG | CAAGCAGAAGACGGCATACGAGATCGATTAGTCTCGTGGGCTCGGAGATG*T |
| rev ATAC-seq primer, Truseq index, 41 GACGAC | CAAGCAGAAGACGGCATACGAGATGTCGTCGTCTCGTGGGCTCGGAGATG*T |

libraries were pooled in equimolar amounts and sequenced on Illumina NextSeq 500. The step-by-step protocol is also provided as Supplemental Data S1.

### TAF-ChIP and conventional ChIP-qPCR

2 μl of TAF as well as conventional ChIP library was used for checking enrichment with various primer pairs (listed in Table 1) on Applied Biosystem ViiA 7 real-time machine using SYBR green reagent (Cat. No. 4367659; Life Technologies).

### Demultiplexing and mapping

Demultiplexing and fastq file conversion were performed using blc2fastq (v.1.8.4). Reads from ChIP-Seq libraries were mapped using bowtie2 (v. 2.2.8) (23) and filtered for uniquely mapped reads. The genome build and annotation used for all *Drosophila* samples was BDGP6 (ENSEMBL release 84). The genome build and annotation used for the K562 samples was hg38 (ENSEMBL release 84).

### Normalization, peak calling, and overlaps

The mapped BAM files were normalized to RPKMs using deepTools, and bigwig coverage files were generated. Peak calling was performed using MACS2 (v 2.1.1-20160309) (24). The peaks were called with the following settings: (i) for *Drosophila* H3K4Me3, macs2 callpeak -t ChIP.bam -c Control.bam -f BAMPE -g dm −q 0.05; (ii) for *Drosophila* H3K9Me3, macs2 callpeak -t ChIP.bam -c Control.bam -f BAMPE -g dm −broad −broad-cutoff 0.05; and (iii) for K562 H3K9Me3 and H3K27Me3, macs2 callpeak -t ChIP.bam -c Control.bam -f BAMPE

-g hs –broad –broad-cutoff 0.05. The resulting peaks were annotated with the ChIPseeker package from Bioconductor, using nearest gene to peak summit as assignment criteria (25). The individual peaks of corresponding modification and approach were merged using bedTools, except ENCODE peak files (26). For ENCODE, owing to high variability between the replicates, we used replicated peaks provided by ENCODE database. After merging, the overlaps were calculated with ChIPpeakAnno package with following commands: (i) for H3K4Me3 in *Drosophila* NSCs, overlap = find OverlapsOfPeaks(ConvK4Me3, TAFK4Me3, maxgap = 100); (ii) for H3K9Me3 in *Drosophila*, overlap = findOverlapsOfPeaks(ConvK9Me3, TAFK9Me3, maxgap = 200); (iii) for H3K27Me3 in K562, overlap = findOverlapsOfPeaks(ENCODEK27Me3, TAFK27Me3, maxgap = 4000); and (iv) for H3K9Me3 in K562; overlap = findOverlapsOfPeaks(ENCODEK9Me3, TAFK9Me3, maxgap = 4000).

## Computational scripts

All the parameters used for computational analysis and detailed scripts are provided in a separate Supplemental Data S2. The heat maps were generated using deepTools (v 3.5.1) (26).

## Accession number

All the ChIP-Seq data generated in this study are submitted to the GEO database (GSE112633).

# Supplementary Information

# Acknowledgements

We thank the Bloomington *Drosophila* Stock Center for the fly lines used in this study. The EMBL Genomics Core Facility (Heidelberg, Germany), especially Vladimir Benes for all the sequencing runs. The IMB FACS facility for the help with sorting of the neuroblast and K562 cells. We thank Jean-Yves Roignant, Guillaume Junion, Joachim Urban, and Prasad Chitke for critically reading the manuscript. This work was supported by the funding Deutsche Forschungsgemeinschaft (DFG) BE 4728 1-1 and 3-1 to C Berger, and grant SI 1991/1-1 to M Silies. The open access funding for publication was provided by University of Mainz. P More, S Albrecht, and A Kulkarni thank the International PhD Programme of the Institute of Molecular Biology, Mainz, for supporting the PhD. The work of F Marini is supported by the German Federal Ministry of Education and Research (BMBF 01EO1003).

## Author Contributions

J Akhtar: conceptualization, formal analysis, investigation, methodology, project administration, and writing—original draft.
P More: formal analysis and writing—review and editing.
S Albrecht: data curation, formal analysis, and writing—review and editing.
F Marini: formal analysis and writing—review and editing.
W Kaiser: dissecting larval brain.
A Kulkarni: dissecting larval brain.
J-F Fontaine: supervision and writing—review and editing.
L Wojnowski: writing—review and editing.
MA Andrade-Navarro: writing—review and editing.
M Silies: writing—original draft.
C Berger: funding acquisition and writing—review and editing.

## Conflict of Interest Statement

The authors declare that they have no conflict of interest.

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
