## [Reviewer comments · Life Science Alliance]

Life Science Alliance

TAF-ChIP: An ultra-low input approach for genome wide chromatin immunoprecipitation assay

Junaid Akhtar, Piyush More, Steffen Albrecht, Federico Marini, Waldemar Kaiser, Apurva Kulkarni, Leszek Wojnowski, Jean-Fred Fontaine, Miguel A. Andrade-Navarro, Marion Silies and Christian Berger

DOI: <https://doi.org/10.26508/lsa.201900318>

Corresponding authors: Junaid Akhtar and Christian Berger, Johannes Gutenberg University

Review Timeline:

Submission Date:	2019-01-24
Editorial Decision:	2019-02-26
Revision Received:	2019-04-20
Editorial Decision:	2019-06-07
Revision Received:	2019-06-13
Accepted:	2019-07-11

Scientific Editor: Andrea Leibfried

Transaction Report:

February 26, 2019

Re: Life Science Alliance manuscript #LSA-2019-00318

Dear Dr. Akhtar,

Thank you for submitting your manuscript entitled "TAF-ChIP: An ultra-low input approach for genome wide chromatin immunoprecipitation assay" to Life Science Alliance. The manuscript was assessed by expert reviewers, whose comments are appended to this letter.

As you will see, the reviewers think that your method will be of value to the field and they provide constructive input on how to further validate the approach to make your manuscript suitable for publication here. We would thus like to invite you to submit a revised version of the manuscript, addressing the reviewers' concerns. Doing so seems straightforward, but please get in touch in case you would like to discuss individual revision points further.

Thank you for this interesting contribution to Life Science Alliance. We are looking forward to receiving your revised manuscript.

Sincerely,

B. MANUSCRIPT ORGANIZATION AND FORMATTING:

Reviewer #1 (Comments to the Authors (Required)):

In the manuscript entitled "TAF-ChIP: an ultra-low input approach for genome wide chromatin immunoprecipitation assay" Akhtar and colleagues present a method to perform ChIP-seq for histone modifications from low cell numbers. The manuscript is very well written and presented and therefore easy to follow. This optimized protocol will be highly relevant for the scientific community. However, some questions need to be addressed in order to control for the accuracy of the protocol. These will be necessary for publication in LSA.

Major comments

1- Method section (TAF-ChIP and library preparation): "the unfragmented chromatin were centrifugated at 14,000 rpm for 10 min at 4C". Are the authors not afraid of losing big chromatin segments during the centrifugation? Is the chromatin getting sonicated a bit from the lysis step?

Have they tried to run an agarose gel from DNA extraction before and after the lysis step to assess this point? Have they quantified DNA before and after the centrifugation step?

2- Same method section: What is the size of the DNA after tagmentation? The authors present a Bioanalyzer profile at the end of the library prep but not after tagmentation (supplementary figure 2A). This would be necessary to assess the distribution of the DNA, if the tagmentation step is optimal and if a lot of DNA is lost during the AMPure purification (such as heterochromatin which is likely less accessible to Tn5).

3- Figure 4C: the metagene profiles for H3K4me3 are vastly different between conventional ChIP and TAF ChIP, specifically at the TSS and TES. This argues that in some instances these two methods are not comparable and give opposite results. The reason(s) for these differences need to be addressed. Can this be due to the fact that the authors use H3 as a control for TAF ChIP and input for conventional ChIP? When comparing to the metagene in figure S4D the TAF H3K4me3 peaks where H3 deeps. Or would this be a biased of tagmentation (more nucleosome free regions therefore more accessible to the enzyme)?

4- In relation to the previous comment, is H3 the right control? It is known that histones are not equally distributed across the genome, it therefore cannot be used as a real input, and could give biased results in some instances.

5- Some ChIP qPCR would be needed to validate the findings, especially when results are conflicting between conventional and TAF ChIP, like in figure 4C.

Minor comments

1- When Encode comparisons are used, are the ChIPs performed with the same antibodies?

Reviewer #2 (Comments to the Authors (Required)):

In this manuscript, Akhtar and colleagues present a further improved version of ChIP-seq enabling epigenetic profiling on very low amounts of starting material. The technology makes use of Tagmentation-assisted fragmentation, preventing the need for sample sonication and integrates library preparation in the sample work-up. The approach is of clear added value to the field. A number of issues do need to be addressed.

1. The authors only provide applicability of the approach with ChIP-seq on histone modifications, which is considerably easier (and intrinsically different, biologically being an intrinsic part of the chromatin) as compared to transcription factors. Therefore, the limitation that the technology as now only been shown on histone modifications should be explicitly mentioned in the title and abstract

2. Figure 2A, B: in the text, it is stated that ENCODE data was used, but this is only the case for figures C, D, E, F. please correct

3. Figure 2G, H. data label for the color scale is lacking.

4. Could the authors also include a heatmap (comparable to 2G, H) for H3K9me3?

5. Pie charts in Figure 3E and 4E: colours are not the same for the 4 different subpanels.

6. Figure 2F and 3F: please show the raw read counts for the different subsets of peaks shared or unique between the methods for H3K27me3 and H3K9me3. Are these really unique, or just subtle different in intensity?

7. Figure 3F, lower panel: over 1 million peaks are found for H3K9me3. This is very high and possibly an artefact induced by the peak caller? Also, since the number of peaks identified for the same

histone modification in Figure 4F are merely 4000-ish... Please check.

minor issues:

1. in the data for Figure 2, only 100 cells were used in the TAF-seq, which is quite an achievement. It would be beneficial to have this explicitly stated in the figure itself.

Reviewer #3 (Comments to the Authors (Required)):

Akhtar et al. describe a new approach to chromatin immunoprecipitation that enables high quality data to be generated from limited amounts of starting material (100s - 1000s of cells). They identify chromatin preparation via sonication as a key limitation of conventional approaches due to variability across machines and potential damage to key epitopes. A second limitation is the use of standard library preparation techniques involving multiple steps including ligation that can limit the sensitivity of the results. The authors attack these limitations employing hyperactive Tn5 to fragment the chromatin while simultaneously inserting primers that allow for immediate amplification of the captured (IP'd) regions. They show histone methylation ChIP-seq data from human cell lines (K562) and Drosophila to demonstrate their approach. Importantly, they compare their results not just to conventional CHIP-seq protocols but also CUT & RUN, a new technique that also avoids standard chromatin prep and enables low cell numbers.

Enabling ChIP-seq to provide reliable data from smaller amounts of input material is an important area to be addressed for scientific community using ChIP. Overall the data in this contribution is compelling and worthy of publication especially as it involves simple steps with commercially available reagents. However, some additional clarifications of the technique need to be added.

1) The authors need to clarify some details of their approach. One would expect that the open regions of chromatin will be highly preferentially tagmented but the amount of Tn5 used appears in TAF-CHIP to be the same as that used in ATAC-seq experiments. This is important since the Tn5 reaction is not catalytic (once the transposase has inserted its loaded oligomer it cannot add any more). For this reason, the total amount of enzyme needs to be at least roughly matched to the amount of input material. The authors mention in the discussion (page 10) where they write "unlike ATAC-Seq where intact cells are tagmented and partial tagmentation is used to study chromatin accessibility...". This is not correct - in ATAC-seq intact nuclei are tagmented to completion with limiting amounts of Tn5.

Was the same amount of Tn5 used in every experiment regardless of cell number? Was a titration of the Tn5 performed to determine the optimum amount? Was bias seen more at the high or low end of the cell numbers examined?

2) Related to the above comment, the authors should provide more detail describing H3 ChIP as a control. This is primarily addressed in Figure S4 which states "H3 TAF-ChIP do not show any visible biases for open chromatin". It is important to compare their data to open chromatin regions in K562 as determined by ATAC-seq not just TSS regions as in Figure S4D. Also they should show genome browser tracks of the control data in Figure S4A.

Minor points:

- CUT&RUN has been shown to be applicable to enhancer marks and transcription factor ChIP-seq.

Is TAF-CHIP? Authors should at least comment.

-What were the parameters used in calling peaks with MACS2, specifically was broad peak calling used for the K27me3 and K9me3 data? This detail should be in the methods.

-Authors should avoid the claims of not employing sonication as they do (fine to say 'limited' or 'low power' etc.).

-They should also not claim their chromatin is not fragmented (show data supporting this statement if it is!) The authors state in the methods section (page 15) "The unfragmented chromatin were centrifuged at 14,000 rpm for 10 min at 4{degree sign}C, and the supernatant was transferred to the tube with blocked and antibody coupled beads." It is unlikely that completely unfragmented chromatin would be soluble and quantitatively available in the supernatant. It is most likely much larger fragments than typically used for CHIP but still fragmented.

We would like to thank the reviewers for their constructive comments, and important feedbacks. Our point by point response to the comments are mentioned below.

Reviewer #1 (Comments to the Authors (Required)):

In the manuscript entitled "TAF-ChIP: an ultra-low input approach for genome wide chromatin immunoprecipitation assay" Akhtar and colleagues present a method to perform ChIP-seq for histone modifications from low cell numbers. The manuscript is very well written and presented and therefore easy to follow. This optimized protocol will be highly relevant for the scientific community. However, some questions need to be addressed in order to control for the accuracy of the protocol. These will be necessary for publication in LSA.

Major comments

1- Method section (TAF-ChIP and library preparation): "the unfragmented chromatin were centrifugated at 14,000 rpm for 10 min at 4C". Are the authors not afraid of losing big chromatin segments during the centrifugation? Is the chromatin getting sonicated a bit from the lysis step? Have they tried to run an agarose gel from DNA extraction before and after the lysis step to assess this point? Have they quantified DNA before and after the centrifugation step?

We would really like to thank the reviewer for bringing this to our notice. This was a major oversight from us, the actual centrifugation speed at this step was 2000g. Now we have corrected this and tallied the method section with our actual protocol hand-out. Also, we are now providing a step-by-step handout of the protocol as an supplementary file that can be useful for the users while carrying out the method. We have also attached a electrophotogram of before and after tagmentation in supplementary figure, as explained below.

2- Same method section: What is the size of the DNA after tagmentation? The authors present a Bioanalyzer profile at the end of the library prep but not after tagmentation (supplementary figure 2A). This would be necessary to assess the distribution of the DNA, if the tagmentation step is optimal and if a lot of DNA is lost during the AMPure purification (such as heterochromatin which is likely less accessible to Tn5).

We provide the bioanalyzer profile before and after tagmentation, in supplementary figure 1A. For the amounts mentioned in the manuscript we found 1 μ l of Tn5 tagmentase (from Illumina) non-limiting, during the method optimization. However, we have titrated the cells to higher numbers while keeping the enzyme amount constant. For generating these profiles three independent tubes were tagmented and pooled to have a profile which could be visualized on Agilent high sensitivity DNA-chip. We avoided PCR amplification, as extension time could be limiting factor and can partially enrich shorter fragments.

To especially address the accessibility issue associated with Tn5, we decided to profile the H3K9Me3 and H3K27Me3 marks. These marks are usually associated with repressed regions and we obtained profiles that looked similar to the one obtained with the conventional ChIP-Seq method. Also, global analysis included in the manuscript support this. In order to further demonstrate it qualitatively we are attaching a genome browser track of H3K9Me3 from *Drosophila* neuroblast, of TAF-ChIP and conventional ChIP-Seq, with zoomed out view of 2R chromosomes. The track clearly shows the enrichment of H3K9Me3 at pericentromeric heterochromatin regions, enriched both in the TAF as well as conventional ChIP-Seq datasets.

Figure: The genome browser track view showing broad enrichment domains of H3K9Me3 in pericentromeric heterochromatin region of chromosome 2R. TAF-ChIP dataset is depicted with green barplots while conventional ChIP-Seq dataset is depicted with red colour.

3- Figure 4C: the metagene profiles for H3K4me3 are vastly different between conventional ChIP and TAF ChIP, specifically at the TSS and TES. This argues that in some instances these two methods are not comparable and give opposite results. The reason(s) for these differences need to be addressed. Can this be due to the fact that the authors use H3 as a

control for TAF ChIP and input for conventional ChIP? When comparing to the metagene in figure S4D the TAF H3K4me3 peaks where H3 deeps. Or would this be a biased of tagmentation (more nucleosome free regions therefore more accessible to the enzyme)?

We completely agree that the profile is different compared to conventional ChIP, especially at the TSS and TES. This slight bias might be indeed due to H3 distribution. We have also used IgG from the same species as a control. We generated metagene profiles using IgG TAF-ChIP as a control, and included them in the revised manuscript in Figure 4C .

4- In relation to the previous comment, is H3 the right control? It is known that histones are not equally distributed across the genome, it therefore cannot be used as a real input, and could give biased results in some instances.

We agree with the reviewer that the H3 profile can be biased. As mentioned before, we have also used IgG as input control. However, with similar immunoprecipitation conditions we have usually struggled to recover enough DNA to produce an equimolar amount of library and have only been able to sequence it to lower depth. This motivated us to use histone H3 as an alternative control for histone ChIPs. Other works have used this successfully [1, 2]. Possible biases in using H3 as control did not impact peak calling for *Drosophila* samples and we obtained similar peaks with or without it. We have added these considerations to the manuscript.

5- Some ChIP qPCR would be needed to validate the findings, especially when results are conflicting between conventional and TAF ChIP, like in figure 4C.

We have performed qPCR to validate the findings as suggested by the reviewer, and included the results in Supplementary figure 3D. The results reveal comparable enrichments at the tested loci, supporting our ChIP-Seq data.

Minor comments

1- When Encode comparisons are used, are the ChIPs performed with the same antibodies?

The antibodies which we have used are not the same as ENCODE. We have used H3K27Me3 and H3K9Me3 antibodies from active motif that we have validated in various qPCR experiments.

Reviewer #2 (Comments to the Authors (Required)):

In this manuscript, Akhtar and colleagues present a further improved version of ChIP-seq enabling epigenetic profiling on very low amounts of starting material. The technology makes use of Tagmentation-assisted fragmentation, preventing the need for sample sonication and integrates library preparation in the sample work-up. The approach is of clear added value to the field. A number of issues do need to be addressed.

1. The authors only provide applicability of the approach with ChIP-seq on histone modifications, which is considerably easier (and intrinsically different, biologically being an intrinsic part of the chromatin) as compared to transcription factors. Therefore, the limitation that the technology as now only been shown on histone modifications should be explicitly mentioned in the title and abstract.

We thank the referee for making this point. Now we have clearly mentioned the application in the abstract. On the other hand, we have been successfully able to immunoprecipitate an RNA modifying enzyme (known to bind chromatin in vertebrates) from a tagged transgenic fly line using antibody against the tag (data not shown).

2. Figure 2A, B: in the text, it is stated that ENCODE data was used, but this is only the case for figures C, D, E, F. please correct .

The ENCODE data is used as reference to generate the PRC and ROC curve, now we have modified the legend accordingly to mention this.

3. Figure 2G, H. data label for the color scale is lacking.

Now the data label is added to the color scale.

4. Could the authors also include a heatmap (comparable to 2G, H) for H3K9me3?

The heatmaps are already available in the supplementary figure 1B and 1D. To make it easier to notice, we have highlighted the relevant samples with colored rectangular boxes.

5. Pie charts in Figure 3E and 4E: colours are not the same for the 4 different subpanels.

This has been modified in the revised manuscript.

6. Figure 2F and 3F: please show the raw read counts for the different subsets of peaks

shared or unique between the methods for H3K27me3 and H3K9me3. Are these really unique, or just subtle different in intensity?

We have generated boxplots showing the distribution of reads over the identified peaks in supplementary figure 1E-1H. To generate these plots we have plotted the distribution of raw reads over the different subsets of peaks, shared or unique, as suggested by the reviewer. As a reference, we have also plotted the raw reads over randomly selected regions of average peak length in respective datasets. We have noticed, especially in K562 datasets, that the reads in unique peaks in either of the approach is indeed slightly higher than the randomly selected regions. This suggests that those peaks are not identified due to FDR thresholding effect implemented in the peak caller software. This is also supported by the ROC curves and PRC curves generated without FDR thresholding in Fig 2A and 2B. The differences in the batch/passage of K562 cells and antibodies used for ENCODE dataset and TAF approach could also be a contributing factor in this variance. Consistent with this, the overlap between peaks was much higher for *Drosophila* samples where identical antibodies and same type of cells (from FACS sorting the same animals) where used for both approaches.

7. Figure 3F, lower panel: over 1 million peaks are found for H3K9me3. This is very high and possibly an artefact induced by the peak caller? Also, since the number of peaks identified for the same histone modification in Figure 4F are merely 4000-ish... Please check.

We would like to thank the reviewer for bringing this to our notice. Unfortunately we used wrong peak files for generating Figure 3F. This has been revised in the manuscript. Now, we have thoroughly checked all our peak files used in the manuscript.

minor issues:

1. in the data for Figure 2, only 100 cells were used in the TAF-seq, which is quite an achievement. It would be beneficial to have this explicitly stated in the figure itself.

The labels in the figure are now modified to clearly highlight the number of cells used in each experiment.

Reviewer #3 (Comments to the Authors (Required)):

Akhtar et al. describe a new approach to chromatin immunoprecipitation that enables high quality data to be generated from limited amounts of starting material (100s - 1000s of cells). They identify chromatin preparation via sonication as a key limitation of conventional

approaches due to variability across machines and potential damage to key epitopes. A second limitation is the use of standard library preparation techniques involving multiple steps including ligation that can limit the sensitivity of the results. The authors attack these limitations employing hyperactive Tn5 to fragment the chromatin while simultaneously inserting primers that allow for immediate amplification of the captured (IP'd) regions. They show histone methylation ChIP-seq data from human cell lines (K562) and Drosophila to demonstrate their approach. Importantly, they compare their results not just to conventional ChIP-seq protocols but also CUT & RUN, a new technique that also avoids standard chromatin prep and enables low cell numbers.

Enabling ChIP-seq to provide reliable data from smaller amounts of input material is an important area to be addressed for scientific community using ChIP. Overall the data in this contribution is compelling and worthy of publication especially as it involves simple steps with commercially available reagents. However, some additional clarifications of the technique need to be added.

1) The authors need to clarify some details of their approach. One would expect that the open regions of chromatin will be highly preferentially tagmented but the amount of Tn5 used appears in TAF-CHIP to be the same as that used in ATAC-seq experiments. This is important since the Tn5 reaction is not catalytic (once the transposase has inserted its loaded oligomer it cannot add any more). For this reason, the total amount of enzyme needs to be at least roughly matched to the amount of input material. The authors mention in the discussion (page 10) where they write "unlike ATAC-Seq where intact cells are tagmented and partial tagmentation is used to study chromatin accessibility...". This is not correct - in ATAC-seq intact nuclei are tagmented to completion with limiting amounts of Tn5. Was the same amount of Tn5 used in every experiment regardless of cell number? Was a titration of the Tn5 performed to determine the optimum amount? Was bias seen more at the high or low end of the cell numbers examined?

We would like to thank the reviewer to pointing this out. Now we have corrected this from "unlike ATAC-Seq where intact cells are tagmented and partial tagmentation is used to study chromatin accessibility..." to "unlike ATAC-Seq where intact nuclei are tagmented and partial tagmentation is used to study chromatin accessibility, our approach tagments after nuclear lysis". When we followed the ATAC-Seq protocols (for an unrelated project) the tagmented and amplified DNA from similar cells had very different profiles, suggesting the effect of nuclear lysis on removing accessibility biases (at least to a greater extent).

As mentioned before, in our optimization experiments for 1000 and 5000 *Drosophila* cells we found the amount described in the manuscript non-limiting. Now, we include a bioanalyzer profile clearly demonstrating this.

2) Related to the above comment, the authors should provide more detail describing H3 CHIP as a control. This is primarily addressed in Figure S4 which states "H3 TAF-CHIP do not show any visible biases for open chromatin". It is important to compare their data to open chromatin regions in K562 as determined by ATAC-seq not just TSS regions as in Figure S4D. Also they should show genome browser tracks of the control data in Figure S4A.

Now we have generated a metagene profile showing the distributions of ATAC-Seq and DNase-Seq from Schmidl C et al (PMID: 26280331), in Figure S4E with different TAF-CHIP Profiles. The read distribution for H3 TAF-CHIP is very different from DNase-Seq and ATAC-Seq datasets, demonstrating the absence of significant biases for open regions. Consistent with the expectation, the K27Me3 and K9Me3 TAF-CHIP datasets also do not show any enrichment for open regions.

Minor points:

- CUT&RUN has been shown to be applicable to enhancer marks and transcription factor CHIP-seq. Is TAF-CHIP? Authors should at least comment.

We have been able to produce Pol II, H3K27Ac and H3K4Me1 datasets from mouse (for an ongoing collaborative project) using TAF-CHIP approach. So if the reviewer meant, using Pol II distribution and histone marks as a hallmark for enhancer signature, then TAF-CHIP can be applicable to interrogate enhancers. As mentioned earlier, we have also produced the profile of an RNA modifying enzyme's chromatin association from transgenically tagged flies. We are providing the unpublished profile along with the control, as a private repository for the purpose of review. This has led us to strongly believe that if the binding is not very limited then the approach can be used to obtain CHIP profiles from tagged animals, or in cases where the antibody is well characterized. The tag alone condition would be an appropriate control for this set-up.

-What were the parameters used in calling peaks with MACS2, specifically was broad peak calling used for the K27me3 and K9me3 data? This detail should be in the methods.

Now all the MACS2 commands used for peak calling are included in the methods section, as well as a supplementary file where all the computational scripts used in the study are

included. Yes, the peaks were called for K27me3 and K9me3 with broad peak calling parameter on.

-Authors should avoid the claims of not employing sonication as they do (fine to say 'limited' or 'low power' etc.).

This has been revised in the manuscript.

-They should also not claim their chromatin is not fragmented (show data supporting this statement if it is!) The authors state in the methods section (page 15) "The unfragmented chromatin were centrifuged at 14,000 rpm for 10 min at 4{degree sign}C, and the supernatant was transferred to the tube with blocked and antibody coupled beads." It is unlikely that completely unfragmented chromatin would be soluble and quantitatively available in the supernatant. It is most likely much larger fragments than typically used for CHIP but still fragmented.

As previously mentioned, the actual centrifugational speed used after nuclear lysis was 2000g for 10 minutes instead of 14,000 rpm erroneously reported. Nonetheless, now we also include the before tagmentation profile along with tagmentation profiles of different cell numbers prior to PCR amplification.

References

1. Bonn, S., *Tissue-specific analysis of chromatin state identifies temporal signatures of enhancer activity during embryonic development*. Nat Genet., 2000. 244(2):148-156.
2. Flensburg, C., *A comparison of control samples for ChIP-seq of histone modifications*. Front. Genet., 2014. 5: 329.

June 7, 2019

Re: Life Science Alliance manuscript #LSA-2019-00318R

Dear Dr. Akhtar,

Thank you for submitting your revised manuscript entitled "TAF-ChIP: An ultra-low input approach for genome wide chromatin immunoprecipitation assay" to Life Science Alliance. We had contacted two of the original reviewers for re-review and have received feedback from one of them. This reviewer evaluated your response to all three original reviewer reports.

As you will see, the reviewer notes that some major concerns of his/her as well as of previous reviewer #3 have not been addressed in the revision, and the reviewer does not support publication of the revised version here. Based on the input received, we concluded that the superiority/value of your method has not been sufficiently demonstrated. As outlined in our previous decision letter, papers are generally considered through only one revision cycle and strong support from the referees on the revised version is needed for acceptance. Given the significant remaining reviewer concerns, we unfortunately cannot offer publication of the manuscript.

We appreciate the effort that has gone into the revisions and regret that the outcome is not more positive.

Sincerely,

Andrea Leibfried, PhD
Executive Editor
Life Science Alliance

Reviewer #1 (Comments to the Authors (Required)):

In this revised manuscript, Akhtar and colleagues have addressed several of the reviewers' comments. While the manuscript has improved and would be of high interest for the scientific community as an alternative method to ChIP-Seq and CUT&RUN, two major concerns raised by reviewers 1 and 3 are still not addressed.

1- The authors can still not claim that their chromatin is not fragmented as they still not show data proving that point. In order to do so they should provide a gel showing the size of the chromatin before and after the lysis step - not only before tagmentation (i.e. after the lysis step). Even after a 2000g centrifugation unfragmented chromatin can be pelleted with debris and lost. Related to that point, the authors did not assess the amount of DNA before and after lysis/centrifugation as suggested.

2- I am still not convinced that H3 is the right control. Why have the authors not tried to use a regular input as a control? I agree with them that IgG should not be IPing a lot of material - hence

hard to use as a control. Besides, the input of DNA is important, IgG assess specificity and cleanness of the protocol. In figure S4A the authors now present H3 tracks. This track should not be presented at the same scale as K4me3; this scale can hide the fact that H3 is not equally distributed. In addition, the authors have not really addressed the question of reviewer 3 related to comparing their data to open chromatin regions other than TSS. What is the difference between TAF H3 in figures S4D and S4E? Why don't these TAF H3 look similar? If we look at TAF H3 in S4D H3 is depleted where the chromatin is more accessible in S4E - which is the concern. This would explain the major difference seen in figure 4C between TAF versus H3 and CHIP Seq versus input - where the latter is depleted where the former is enriched right before the TSS and at TES.

Thank you very much for your email. As you can imagine, we are very disappointed by your decision.

I would be grateful if we could discuss this by phone or email. Overall, the comments of Reviewer #1 seemed very positive, as apparent from his judgement that our work “would be of high interest for the scientific community”. We appreciate that Life Science Alliance is taking its effort to make the reviewing process straightforward serious, allowing only one round of major revisions. However, it did not go unnoticed to us that some LSA articles were allowed another round of revisions. While the reviewer still raised two remaining points of concern, we think that these could be easily addressed. We took the liberty to directly address the remaining concerns of the reviewer below. Accordingly, we have also modified our manuscript which I am submitting with this letter.

We would very much appreciate if (i) you could critically evaluate our response, (ii) send it back to Reviewer #1, or (iii) wait to include a response of the other two reviewers. In fact, the two other reviewers were already very positive upon the first submission, where Reviewer #2 was only listing minor comments, and stated that “the approach is of clear added value to the field”. Reviewer #3 also was very positive upon initial submission and said: “Overall the data in this contribution is compelling and worthy of publication especially as it involves simple steps with commercially available reagents” and just asked for “additional clarifications”. We therefore hope that you agree that a rejection at this stage would likely not be in agreement with the opinion of any of the three reviewers. We would appreciate to at least include their feedback at this stage.

Again, I'd be more than happy to discuss this by phone at your convenience.

Thank you very much for your time and consideration.

Best regards,

Junaid Akhtar

Reviewer #1 (Comments to the Authors (Required)):

In this revised manuscript, Akhtar and colleagues have addressed several of the reviewers' comments. While the manuscript has improved and would be of high interest for the scientific community as an alternative method to ChIP-Seq and CUT&RUN, two major concerns raised by reviewers 1 and 3 are still not addressed.

We thank the reviewer for this very positive evaluation of our work. We are going to address the remaining concerns in detail below.

1- The authors can still not claim that their chromatin is not fragmented as they still not show data proving that point. In order to do so they should provide a gel showing the size of the chromatin before and after the lysis step - not only before tagmentation (i.e. after the lysis step). Even after a 2000g centrifugation unfragmented chromatin can be pelleted with debris and lost. Related to that point, the authors did not assess the amount of DNA before and after lysis/centrifugation as suggested.

We would like to thank the reviewer's careful evaluation of the method which has helped us in improving our manuscript. The reviewer is asking for further evidence addressing chromatin fragmentation, and is suggesting an analysis on a gel. The amount of DNA present in a TAF-experiment is ~153 pg (0.152pg DNA / cell), which unfortunately is not amenable to standard gel-based size separation techniques. An alternative method to gel-based size separation approach is the Agilent Bioanalyzer, widely used to assess shearing efficiency for even conventional ChIP-Seq approach. Due to this, we resorted to this approach for size estimation, and as shown in the attached profile there wasn't any indication of fragmentation. The before tagmentation profile is in principle the profile after the lysis step, as there are no intermediate steps involved which can fragment the chromatin. Also we have not recovered any visible pellet or detectable amount of DNA (left in the tube after centrifugation), thus cannot comment on the loss of the DNA during centrifugation.

Independent of this argument, as mentioned in the handout of the protocol (providing step by step details of the method), we have also performed the method omitting this step altogether without compromising the outcome. The cellular debris in this case is dealt with washes after tagmentation, bypassing the need for centrifugation. We now explicitly mention this in the method section of the manuscript.

Lastly, we have toned down this claim, as partial fragmentation (if present, though not detectable) does not compromise the successful implementation of this approach for a limited amount of starting material. We have benchmarked our data diligently against the contemporary alternative approaches and demonstrated the applicability/user friendliness of our approach.

2- I am still not convinced that H3 is the right control. Why have the authors not tried to use a regular input as a control? I agree with them that IgG should not be IPing a lot of material - hence hard to use as a control. Besides, the input of DNA is important, IgG assess specificity and cleanness of the protocol.

We would like to thank the reviewer for her/his valuable input. Unfortunately, it is not possible to obtain a conventional input control. A regular input in conventional ChIP-Seq comprises of an aliquot of sonicated material from the IP buffer, however in this case the aliquot would be (A) very limited in amount and (B) predominantly unfragmented

Lastly, unlike other contemporary low amount ChIP-Seq (including CUT & RUN) we benchmarked our data with input normalization. At the same time, we still identified overlapping peaks without using input control as stated in our earlier response.

In figure S4A the authors now present H3 tracks. This track should not be presented at the same scale as K4me3; this scale can hide the fact that H3 is not equally distributed.

The reviewer is right that the tracks in Fig. S4A can potentially hide a non-uniform distribution.

We are now providing a figure to the reviewer (Figure R1) showing that the signal for H3 still looks very distinct from H3K4me3 distribution. The figure below was produced by grouping the H3K4me3 tracks together and letting the genome browser software determine the scale (IGV). Similarly, the two replicates of H3 TAF-ChIP were grouped together and the scale was automatically determined by the software. We now have replaced Figure S4A with this representation. Additionally, we provide a separate file with genome browser track view of HA tagged RNA modifying enzyme's chromatin binding profile along with the respective tag alone control, for the purpose of this review.

Figure R1: The genome browser track view showing enrichment of H3K4me3 mark using TAF-ChIP approach from 1000 and 5000 *Drosophila* neuroblast (DMNSCs), shown in yellow. Below is also shown H3 TAF-ChIP enrichment profile with 1000 DMNSCs. The scales were auto adjusted by the genome browser software IGV (Broad Institute).

In addition, the authors have not really addressed the question of reviewer 3 related to comparing their data to open chromatin regions other than TSS. What is the difference between TAF H3 in figures S4D and S4E? Why don't these TAF H3 look similar? If we look at TAF H3 in S4D H3 is depleted where the chromatin is more accessible in S4E - which is the concern. This would explain the major difference seen in figure 4C between TAF versus H3 and ChIP Seq versus input - where the latter is depleted where the former is enriched right before the TSS and at TES.

The H3 Profile shown in figure S4D is derived from *Drosophila* neuroblast, while the H3 TAF-ChIP profile shown in figure S4E is from 100 K562 cells. We apologize to the reviewers as it was not explicitly mentioned in the legend of Fig S4D. The legend now reads "Metagene profiles of H3 and H3K4Me3 from 1000 sorted *Drosophila* NSCs with standard error to the mean for all genes.....". The profiles can be different as they are from two different organisms and with a size of 150 Mbs, the *Drosophila* genome is much more compact than the human genome. Also, potentially different H3 distributions might result in different profiles, and thus cannot be directly compared. We again apologize for causing this misunderstanding.

We also believe that we have addressed the comment from Reviewer 3 in Fig S4E. The profile shown is exactly a re-run of the ATAC-Seq and DNase-Seq datasets with H3 TAF-ChIP dataset, as shown in supplementary figure S5C (PMID:26280331). We did not specifically look at the hypersensitive sites as our initial analyses clearly indicated that H3 TAF does not show

any enrichment at open regions like in ATAC-Seq and DNase-Seq when summarized over all genes.

We hope that together, these replies have clarified all remaining points.

Best regards,

Junaid Akhtar

I am sorry for the delay in getting back to you. We felt that your arguments were likely to receive support from the reviewer, but wanted to involve the reviewer at this stage for an expert assessment. I therefore asked reviewer #1 for feedback on your appeal note. I have now heard back from the reviewer and I am happy to say that the reviewer appreciated your arguments and thinks that they address the remaining concerns. We are thus happy to move forward with your paper here.

Please log into our submission system to upload final files. Importantly, please make sure to fill in all fields within the submission system and to have the correct author order. Please list in your manuscript file 10 authors et al. in the reference list. Please also add a callout to figure SFig3A in the manuscript text. The manuscript text needs to get provided as a word docx document, please. I think it would be furthermore nice to include the 'detailed protocol' directly in the methods section of the paper. This is not an issue length-wise.

July 11, 2019

RE: Life Science Alliance Manuscript #LSA-2019-00318RR-A

Dear Dr. Akhtar,

Thank you for submitting your Methods entitled "TAF-ChIP: An ultra-low input approach for genome wide chromatin immunoprecipitation assay". It is a pleasure to let you know that your manuscript is now accepted for publication in Life Science Alliance. Congratulations on this interesting work.

DISTRIBUTION OF MATERIALS:

Again, congratulations on a very nice paper. I hope you found the review process to be constructive and are pleased with how the manuscript was handled editorially. We look forward to future exciting submissions from your lab.

Sincerely,

Editor Life Science Alliance
Scientific Editor
Life Science Alliance